# Shrews Under-Represented in Fruit Farms and Homesteads

**DOI:** 10.3390/ani13061028

**Published:** 2023-03-11

**Authors:** Linas Balčiauskas, Vitalijus Stirkė, Andrius Garbaras, Laima Balčiauskienė

**Affiliations:** 1Nature Research Centre, Akademijos 2, 08412 Vilnius, Lithuania; 2General Jonas Žemaitis Military Academy of Lithuania, Šilo str. 5A, 10322 Vilnius, Lithuania

**Keywords:** insectivores, Soricidae, fruit orchards, berry plantations, homesteads, kitchen gardens, diet

## Abstract

**Simple Summary:**

Between 2018 and 2022, we surveyed small mammals at 23 sites in Lithuania—meadows, commercial orchards and berry farms, kitchen gardens, homesteads and farms—with the aim to assess the proportion of shrews in the community and their diet using stable isotope analysis. We found that in these natural, agricultural, and commensal habitats, common (*Sorex araneus*) and pygmy *(Sorex minutus*) shrews were under-represented—having a proportion of 3.1%, less than a half that of the long-term average in the country. The diet of these two species was similar in both agricultural and commensal habitats. On farms and in orchards with intensive farming, there were no catches of shrews. Contamination by plant protection products and a lack of invertebrates, which are the main food of shrews, may be factors limiting their numbers in the agriculturally managed habitats. Two species of water shrews, *Neomys fodiens* and *Neomys anomalus*, were found for the first time in homesteads, including in outbuildings, and their diet requires further investigation.

**Abstract:**

Shrews are a less studied group of small mammals than rodents. Between 2018 and 2022, we surveyed 23 sites in Lithuania, including natural and anthropogenic habitats, with the aim to assess the proportion of Soricidae in small mammal communities and their diet based on stable isotope analysis. The average representation of Soricidae was 3.1%, about half the long-term average in other habitats in the country. The highest proportions were in meadows and farmsteads, at 4.9% and 5.0% respectively. Shrews were not trapped on farms or in young orchards, and their relative abundance was very low in intensively managed orchards (0.006 individuals per 100 trap days). *Neomys fodiens* and *N. anomalus* were unexpectedly found in homesteads, including in outbuildings. *Sorex araneus* and *S. minutus* had similar diets. The trophic carbon/nitrogen discrimination factor between invertebrates and shrew hair was 2.74‰/3.98‰ for *S. araneus*, 1.90‰/3.78‰ for *S. minutus* in the orchards. The diet of *N. fodiens* and *N. anomalus* at the homesteads requires further investigation. We propose that the under-abundance of shrews may be due to contamination by plant protection products and a lack of invertebrates under intensive agricultural practices.

## 1. Introduction

The family Soricidae is divided into two subfamilies, Soricinae and Crocidurinae [1], the latter not currently found in Lithuania. With more than 370 species, the family Soricidae is a very diverse family of mammals [2], exhibiting genetic diversity [3,4]. The family Soricidae originated in Eurasia [5] and is still very abundant in boreal forests [6]. Moisture is thought to be a key factor in the evolution of the Soricidae [5] and is of great importance for the ecology of shrews [7,8]. According to Sheftel and Hanska [6], wet and vegetation-rich habitats in the Eurasian boreal forests are characterized by the highest abundance and species richness of shrew communities.

Four shrew species occur in Lithuania: the common shrew (*Sorex araneus*), the pygmy shrew (*S. minutus*), the water shrew (*Neomys fodiens*) [9], and the Mediterranean water shrew (*N. anomalus*), the last one identified only in 2012 [10]. As for the habitat preferences, *S. araneus* is more common in meadows, the banks of water reservoirs, reedbeds, old gardens, parks, forest edges, swamps, wet leaf litter, and forests, but is less common in dry forests. *S. minutus* is mostly found in wetland biotopes (marshy meadows, reedbeds, raised bogs, fens), but also in meadows, forest edges and forests. *N. fodiens* lives at the edges of various water bodies, covered by trees, shrubs, and dense grasses, and is encountered in biotopes adjacent to water bodies (reedbeds, meadows) and occasionally in forests and meadows farther from water [11]. Until 2020, all *N. anomalus* in Lithuania had been trapped near water, in reedbeds, or in sedge habitats [10]. This is consistent with the habitats investigated by other authors, notably overgrown water banks, marshes, and peat bogs [12,13,14,15].

Data on the relative abundances of shrews and the proportions of their species in small mammal communities are heterogeneous. In the Carpathian Mountains, *S. araneus* accounted for 0.8%, and all insectivores for only 1.3%, of the total number of small mammal individuals trapped [16]. This contrasted with the work of Benedek et al. [17], who reported that the proportion of all insectivorous species in the Carpathians was 27.4%, while *S. araneus* alone was 23.6%. This is also much higher than the 8–15% reported by Bryja and Rehak [18]. According to Baláz and Ambros [19], various shrew species differ in their preferences of forest type and altitude in the Carpathian Mountains.

In Finland, the relative density of *S. araneus* was 0–5 individuals per 100 trap nights for most of a 20-year period, occasionally exceeding 10 ind. per 100 trap nights in autumn during some years [20]. Similar densities of 0.6–3.7 ind. per 100 trap nights for *S. araneus* and much lower densities of 0.03–1.6 ind. per 100 trap nights for *S. minutus* were recorded in forests of Fennoscandia two decades later [21]. In other habitats, the abundance of shrews may differ, e.g., *S. minutus* can dominate in peatbogs. In Poland, this species accounted for 26.4% of all trapped individuals, while all Soricidae species as a whole accounted for 43.1% [22].

In Lithuania, *S. araneus* has been recorded as dominant in the tawny owl (*Strix aluco*) diet: in 1999, in three forests in the central part of the country, shrews accounted for 40.7%, 38.3%, and 46.4%, respectively, while species proportion in these forests in 2000 was 26.7% [23]. A figure of 26.7% was also recorded in a protected wetland site in the north of the country in 2000.

Long-term data on the shrew proportion in small mammal communities are also available from China [24] and several European countries [25,26,27]. A decline in *S. araneus* abundance was recorded in the Czech Republic between 1991 and 2015, with *S. araneus* and *S. minutus* accounting for an average of 8.1% and 1.3% of small mammals, with a maximum 23.0% and 3.7%, respectively, in young spruce stands [26]. Over the last 60 years, significant changes in the shrew communities have also been recorded in the Republic of Moldova [25]: *S. araneus* remains the most abundant species, while *Neomys* has declined considerably, and *Crocidura* has increased. Between 1975 and 2022, the average proportion of *S. araneus* in Lithuania was 10.0%, while that of *S. minutus* was 3.3%. Both species were declining [27]. On average, the proportion of Soricidae in the diet of *S. aluco* between 1999 and 2005 was similar, with *S. araneus* accounting for 7.2%, *S. minutus* 4.5%, and *N. fodiens* 0.1% of the total number of small mammals hunted [28].

Agricultural and anthropogenic habitats may have different compositions of small mammals [29,30,31]. In an Italian agricultural zone (maize, corn, and sunflower fields), *S. araneus* was one of the two main species of prey of barn owls (*Tyto alba*), accounting for about 10% of all small mammals preyed upon, with no habitat preference. *Crocidura* shrews (21%), however, were associated with open and cultivated habitats, while *Neomys* shrews were preyed upon along canals and ditches [32]. Elsewhere, the Greater white-toothed shrew (*Crocidura russula*) accounted for 16.1% of the small mammals recorded in olive groves in Portugal [33], whereas in Serbia, in a mosaic of clover, maize, wheat, and soybean fields, all Soricidae accounted for 23.6% of the prey of owls, of which *S. araneus* accounted for 5.19% and *S. minutus* for 0.86% [34]. In the suburbs of Warsaw (Poland), *S. araneus* accounted for 4.9% and *S. minutus* for 0.2% of the prey of *S. aluco* [35].

Diet composition and resource overlap among species is key to understanding ecological communities, including the adaptation to different environments [36]. Differences in the diet of *S. araneus* and *S. minutus* in grassland habitats have been shown to reduce competition between these species, as shown by studies of the digestive tract contents [37]. In contrast, an almost complete overlap of trophic niches between these species was found in a mountainous area with limited food supply, suggesting competition [38]. In marshland in the Bialowieza Forest in eastern Poland, abundant invertebrate prey allowed four species of Soricidae to coexist. In this case, the diets of *S. araneus, S. minutus, N. fodiens*, and *N. anomalus* overlapped considerably, despite signs of prey selectivity [39]. Using isotopic analysis, an overlapping diet was also found between *S. araneus* and *S. minutus* in a resource-rich floodplain meadow in western Lithuania [40]. The importance of abundant invertebrates as key micro-environmental features for shrews [41,42] was thus confirmed.

Insectivores are not listed as pests; therefore, they are rarely monitored [43]. As with other mammals, climate change [44] is an important factor influencing the distribution and biology of shrews [45,46]. In agricultural habitats, shrews are very important as model species, as they are more likely to be contaminated by various compounds used as insecticides, plant protection products, etc., and, consequently, they show higher contamination levels [47]. Shrews also respond positively to habitat protection measures [17,48] and can therefore demonstrate the effectiveness of the measures.

The aim of this study was to assess the proportions of Soricidae in small mammal communities in meadows, commercial fruit and berry farms, and commensal habitats (homesteads, kitchen gardens, and farms) of Lithuania, as well as their trophic niche, as determined by stable isotope analyses of their hair.

## 2. Materials and Methods

### 2.1. Study Sites

Between 2018 and 2022, small mammals were surveyed at 23 sites in various regions of Lithuania: 10 commercial apple orchards (AO), two plum orchards (PO), three raspberry plantations (RP), three currant plantations (CP), one highbush blueberry plantation (B), two homesteads (H), one kitchen garden (KG), and two farms (F) (Figure 1). The average size of apple orchards was 63.7 ha, of the plum orchards was 0.81 ha, of the currant plantations was 22.0 ha, of the raspberry plantations was 2.3 ha, and of the blueberry plantation was 3.80 ha.

Agricultural study sites were old fruit orchards (AO2–9), middle-aged fruit orchards, and berry plantations (AO10; PO2; CP1–3; RP1, 2; B1), young orchards, and plantations (AO1, 6; PO1; RP3). The intensity of on-site agricultural measures was also different, from high (AO1, 2, 4, 6, 8, 10; RP2, 3; B1), to medium (AO3; PO1; CP3; RP2), and to low (AO5, 7, 9; PO2; CP1, 2; RP1). We characterized three levels of intensity depending on soil scarification, grass mowing, mulching of the plant interlines, and the usage of rodenticides and plant protection agents. Sites with only grass mowing once or several times per season were classified as low intensity, while usage of two measures from those listed above, once or twice per season, was defined as medium intensity. The application of several measures or frequent application of two measures per season was defined as high intensity. The nearest non-agricultural habitat was used as a control in every study site, these being mowed or unmowed meadows, or forest (Table A1).

The kitchen garden was characterized by a limited number of buildings and a limited period of availability of human food. Both homesteads had a variety of buildings, some of which contained human food that was readily available for most of the year. The farms had the most complex building structures and unlimited access to human food and fodder for livestock, poultry, or rabbits throughout the year.

### 2.2. Small Mammal Trapping

In orchards and plantations, small mammals were snap-trapped using a standard method—rows of 25 traps at 5 m spacing, set for three days and checked once per day in the morning. Depending on the size of the orchard, two to four trapping lines were set, while in the meadows, the number of lines was one or two.

Inside the building structures of the kitchen garden, homesteadss and farms, small mammals were trapped opportunistically using 5 to 20 traps. Opportunistic trapping was also performed around all accessible buildings. Thus, the term “commensal habitats” was used sensu lato. More details of trapping are given in [49,50].

In 2018–2021, we trapped small mammals in the orchards and their control meadows twice a year: in June and September–October. In 2022, small mammals were trapped in the orchards only in autumn. Trapping effort in 2018–2022 amounted to 36,978 trap days. Details regarding the trapping effort in orchards and plantations are given in Table A2. Differences in trapping efforts were inevitable due to differences in habitat availability, but, as shown in our previous publications [49,50,51], they did not affect the trappability of the more abundant species and the diversity of the small mammal community. This was tested using species accumulation curves (Figure A1) under different trapping efforts. Using long-term data, it was shown *S. araneus* was the fourth-most abundant species in Lithuania [27]. The species accumulation curves show that it should be trapped with the minimum trapping effort, or a very low number of trapped individuals (Figure A1); therefore the absence of this species cannot be related to under-trapping.

Trapping on the homesteads, farms, and kitchen gardens took place throughout the year, with at least several trapping sessions each season. The trapping efforts on farm F1 amounted to 1530 trapping days and on homestead H2, to 210 trapping days. On homestead H1, the trapping effort in 2019–2020 was 1480 trapping days. In the other commensal habitats, the precise trapping effort was not calculated due to the use of the opportunistic trapping method.

Therefore, the relative abundance of small mammals, including shrews, expressed as the number of individuals per 100 trap days, was not always available for habitats other than orchards.

Trapped shrews were identified by external features, such as the base of the tail and the teeth in the *Sorex* species, using an identification key [9], and the hairy tail keel in the *Neomys* species [10].

### 2.3. Stable Isotope Analysis

In addition to stomach analysis, pellet analysis and laboratory feeding trials were used earlier to investigate shrew diets [37,38,39], and stable carbon and nitrogen isotope signatures in the hair may be used as a diet proxy [40]. Isotopic niches are substitutes for trophic or dietary niches, with a higher ratio of nitrogen (^15^N/^14^N) indicating consumption of animal foods [51].

We analyzed carbon and nitrogen isotopic ratios in the hair of shrews, using an elemental analyzer (Flash EA1112) coupled to an isotope ratio mass spectrometer (Thermo Delta V Advantage) via a ConFlo III interface. About 5 mm of hair was clipped from the back of individuals between the shoulders. Before analysis, dirty hair samples were washed in deionized water and methanol, then desiccated. Dry samples were weighted (0.5–1 mg) into tin capsules and stored in the sample plate. The stable isotope (^13^C/^12^C and ^15^N/^14^N) ratios were expressed relative to international standards, Vienna Pee Dee Belemnite, and atmospheric air, respectively. More details of sample preparation and analyses are given in [40,51].

Prior to preparation and analysis, samples (*n* = 10) of small mollusks and arthropods from orchards were stored in a freezer at below –20 °C. After drying in an oven at 60 °C to a constant weight for 24–48 h, invertebrate samples were homogenized to a fine powder, using mortar and pestle and a Retsch mixer mill MM 400 [51].

### 2.4. Data Analyses

We estimated the proportion of Soricidae and the two most abundant species, *S. araneus* and *S. minutus*, in the total number of small mammals trapped (Table 1), presenting it as mean and 95% CI for each habitat type. Differences in proportions were assessed using the G test from an online calculator [52].

The *δ*^13^C and *δ*^15^N values of the hair samples were expressed as the arithmetic mean ± 1 SE and the range, from the minimum to maximum observed value. The isotopic values of species and trophic groups, including those with sample size *n* < 5, were visualized in isotopic biplots.

We used ANOVA to determine the influence of the habitat, the age of the orchard, and the intensity of the agricultural measures on the dependent parameters: the relative abundance and the hair *δ*^15^N and *δ*^13^C values in *S. araneus* and *S. minutus.* Differences between groups were evaluated with the post-hoc Tukey’s test, and pairwise comparisons were completed using the Student’s *t* test. The normality of the distributions of the hair *δ*^15^N and *δ*^13^C values were tested using the Kolmogorov–Smirnov’s D test online [53]. Homogeneity of variances of the hair *δ*^15^N and *δ*^13^C values in *S. araneus* and *S. minutus* was assessed using the Levene test (Table A3).

The minimum confidence level was set as *p <* 0.05. However, the small sample size of two *Neomys* species may result in a low power for statistical analysis. Calculations were performed in Statistica for Windows, version 6.0 (StatSoft, Inc., Tulsa, OK, USA); biplots were drawn in SigmaPlot ver. 12.5 (Systat Software Inc., San Jose, CA, USA).

## 3. Results

### 3.1. Shrew Proportions in Different Small Mammal Communities

During the study, 3141 small mammal individuals representing 14 species were trapped. Of these, 96 were shrews: 65 *S. araneus*, 26 S. *minutus*, three *N. fodiens*, and two *N. anomalus*. The proportions of Soricidae varied between habitats (G = 37.14, *p <* 0.001), with the highest proportion in the control habitats and homesteads. Shrews were not trapped on farms (Table 1). Of all Soricidae species, the highest number of shrews were trapped in the control habitats, 47.7% (CI = 36.8–58.3%), with lower numbers in apple orchards, 15.5% (CI = 7.3–25.8%), then raspberry plantations, 13.8% (CI = 6.1–23.7%), and currant plantations, 10.3% (CI = 3.9–19.3%).

The relative abundance of all species was low, with *S. araneus* being most abundant in meadows, followed by berry plantations. *S. minutus* was most abundant in meadows (Table 1). The highest relative abundance of *S. araneus* and *S. minutus* was recorded in the control meadows, at 2.67 and 4.00 ind. per 100 trap days, respectively.

Agricultural treatment intensity had no effect on the relative abundance of *S. minutus* (ANOVA, F_2,322_ = 0.92, *p* = 0.43) and a weak effect on the relative abundance of *S. araneus* (F_2,322_ = 3.03, *p <* 0.05). The relative abundance of the latter species was 0.006 ind. per 100 trap days in intensively managed crops, 0.09 in moderately managed crops, 0.19 in low-managed crops, and 0.28 ind. per 100 trap days in control habitats, with a difference of more than ten times.

The relative abundance of *S. araneus* and *S. minutus* was not affected by the age of the orchard or plantation (F = 0.96, *p* = 0.46); however, in the young crops, shrews were not trapped at all.

### 3.2. Stable Isotope Ratios of Shrews

Statistics for the stable isotope ratios of the insectivorous species in agricultural and commensal habitats are given in Table 2.

In the commercial orchards, the trophic niche of *S. araneus* was wider than that of *S. minutus*, with a range of *δ*^13^C values of 1.9-fold, and *δ*^15^N values of 1.5-fold. The two species were fully separated according *δ*^13^C and did not differ according *δ*^15^N (Figure 2). The *δ*^15^N value of *N. fodiens* was about 2.4 times higher than those of *Sorex* shrews.

We calculated the trophic discrimination factor, TDF, of invertebrates (the most likely food for shrews) and the shrew hairs (Figure 2). The TDF of carbon and nitrogen was 2.74‰/3.98‰ for *S. araneus*, 1.90‰/3.78‰ for *S. minutus*, and 3.03‰/13.50‰ for the single *N. fodiens* individual in the orchards. The TDF of nitrogen in the latter species indicates that *N. fodiens* can use not only invertebrates, but also other food sources of animal origin.

In the commensal habitats, namely the kitchen garden and the homestead, the four Soricidae species did not exhibit a distinct separation in dietary space (Table 2), but the number of trapped individuals was minimal. Lower *δ*^15^N values were observed for both *Neomys* species compared to those of *S. araneus* and *S. minutus*; however, a larger sample is required for statistical analysis. In Lithuania, *N. fodiens* and *N. anomalus* were trapped for the first time in commensal habitats such as outbuildings.

Compared with invertebrates, the TDF of carbon and nitrogen in shrews from commensal habitats was similar to those from commercial orchards: for *S. araneus* (2.73‰/3.70‰) and for *S. minutus* (1.63‰/3.88‰). In *N. fodiens*, the TDF was 2.16‰/2.09‰, and in *N. fodiens*, it was almost the same, 2.42‰/2.09‰.

### 3.3. Trophic Position of Insectivores in Relation to Other Groups of Small Mammals

In commercial orchards, the positions of the small mammal trophic groups were well separated (Figure 3a) in terms of both *δ*^13^C (F = 184.3, *p <* 0.001) and *δ*^15^N (F = 28.8, *p <* 0.001).

Four herbivore species, the common vole (*Microtus arvalis*)*,* short-tailed vole (*M. agrestis*)*,* root vole (*M. oeconomus*), and water vole (*Arvicola amphibius*), had the lowest *δ*^13^C and *δ*^15^N values, which were different from all other trophic groups (HSD, *p <* 0.001). The three granivore species, the striped field mouse (*Apodemus agrarius*)*,* yellow-necked mouse (*A. flavicollis*), and harvest mouse (*M. minutus*), showed similar *δ*^13^C values, with their mean different from other trophic groups at *p <* 0.001. In terms of mean *δ*^15^N, they exhibited higher values than did herbivores (*p <* 0.001) and lower than did the insectivores (*p <* 0.05). Insectivores and omnivores, these being the bank vole (*Clethrionomys glareolus*)*,* house mouse (*Mus musculus*) and Northern birch mouse (*S. betulina*), did not differ in terms of either *δ*^13^C or *δ*^15^N means (Figure 3a).

The separation of small mammal trophic groups by *δ*^13^C (F = 16.7, *p <* 0.001) and *δ*^15^N (F = 34.1, *p <* 0.001) was also significant in commensal habitats (Figure 3b). However, intergroup differences were not as pronounced as those in orchards. In commensal habitats, insectivores had higher mean *δ*^13^C values than herbivores (HSD, *p <* 0.01), but did not differ from those of granivores and omnivores.

The mean *δ*^15^N value of insectivores was only higher than that of granivores (HSD, *p <* 0.005). Other small mammal trophic groups were heterogeneous, with herbivorous *M. arvalis* and omnivorous *C. glareolus* having higher *δ*^15^N values than *Neomys* shrews. The granivore *A. agrarius* was at the same level as both *Sorex* species, while the omnivore *M. musculus* was characterized by the highest *δ*^15^N (Figure 3b).

## 4. Discussion

The results of the 2018–2022 small mammal trapping indicate that shrews were an under-represented group in the agricultural and commensal habitats of Lithuania, and their proportion was approximately half the long-term average proportion of Soricidae in other habitats [27]. However, two *Neomys* species were unexpectedly trapped in commensal habitats, including outbuildings far from the nearest water source.

Despite some data on shrews in agricultural and anthropogenic habitats [29,30,31,32,33,34,35], agro-ecological studies are mostly limited to rodent communities [54,55,56,57,58,59,60]. Studies on insectivores in anthropogenic habitats are mostly limited to birds and bats [61,62,63,64,65,66], and the same applies to agricultural areas [67,68,69,70]. As such, the problem is that in some cases, there is no reference point against which to compare recent data [71]. However, insectivores may be important as vectors of pathogens [72,73,74], including *Yersinia enterocolitica* recently identified in *S. araneus* from Great Britain [75]. Therefore, our study is of value, as we provide long-term primary data on insectivores in the country for comparison purposes [27], although knowledge of the pathogen situation is still limited [76].

As for shrews living in commensal habitats, this group can be expected to be under-represented, as this is a general problem of simplified diversity, especially in urban areas [77]. The hostility of anthropogenic environments, such as urban habitats, is compensated by changes in Soricidae behavior, such as increased boldness and high individual variation in aggression [78]. The presence of suitable habitats is of primary importance in the anthropization gradient, but these habitats in urbanized areas create opportunities for contact and interaction between humans or domestic animals with wild animals [79]. Shrews can be found in very unusual habitats, including airport fields: at Chisinau airport in Moldova, one shrew species was recorded from three species found in the adjacent area [80].

We propose the hypothesis that one of the reasons for the low numbers of shrews in agricultural habitats is the biomagnification of pollutants. Shrews have a higher contamination rate than mice [47], and these authors argue that the effectiveness of so-called organic or ecological farming to avoid insectivore pollution is limited. This is partly contradicted by Pelosi et al. [81], who found higher concentrations of pesticide residues in the soil and earthworms of recently treated areas. The higher contamination of shrews than rodents is based on their position in the food chain, and even new “safe” insecticides concentrate in shrews [82]. This is consistent with general patterns of terrestrial vertebrate exposure to pollutants [83]. We will not speculate further on this point, but in our study, in the orchards with the most intensive agricultural treatments (including the use of plant protection products), shrews were not trapped. The susceptibility of shrews to pesticide exposure, “that can be oral via direct consumption and watering or grooming, trophic transfer, inhalation, and/or dermal contact” [84], justified the suitability of shrews as a focal species for risk assessment of plant protection products.

Another reason for their low numbers may be the limited trophic resources available to shrews, these absent due to agricultural treatment in orchards, berry plantations, and homesteads. Such agricultural activity was not present in the other commensal habitats. The diet of shrews is based exclusively on invertebrates [37,39,41,42]. *S. araneus* mainly consumes coleopterans, insect larvae, araneids, opilionids, and isopods [85], whereas araneids, lumbricids, and coleopterans are also common foods for *S. minutus* [38]. These prey groups should be affected in orchards, but are unlikely to be affected in commensal habitats. Recent observations of *S. minutus* in commensal habitats (roof cavities) have been attributed to possible avoidance of harsher climates, resulting in a change in diet [86]. In our study, *S. araneus* and *S. minutus* did not differ in stable isotope values between commensal and agro-habitats (see Figure 2).

With regard to the diet of *Neomys* shrews, there are differences in the prey related to the hunting method, as only *N. fodiens* can hunt underwater [87]. As all of our trappings for both *N. fodiens* and *N. anomalus* were in atypical agricultural or commensal habitats, the dietary characteristics are unknown. Two possible reasons for the high *δ*^15^N value in *N. fodiens* from orchards can be suggested as the influence of fertilization or preying/scavenging on vertebrate food. Occasional vertebrates in the diet of *N. fodiens* have been previously recorded [88]. An increase in *δ*^15^N values in the hair of small mammals was detected under the influence of guano from a colony of great cormorants, *Phalacrocorax carbo* [89]. Therefore, both factors are possible. It is also known that pesticide use can alter the diets of shrews and rodents [90].

The dispersal of shrews between control areas and orchards is possible, as the migration distance of *S. minutus* ranges from 475 to 2570 m [91]. The migration distance of *S. araneus* is unknown, but it has a very limited home range of about 500 sq. m and an activity radius of 13 m [92]. Therefore, the diet of the shrews in our study probably corresponded to the habitat in which they were trapped.

Finally, we suggest that there was no possible influence of removal trapping on shrew presence and abundance sensu Sullivan et al. [93]. Long-term trapping of similar intensity has been used in several studies in Lithuania and has not shown any resultant significant changes in the proportions of any small mammal species, including shrews [27,40,94,95]. Therefore, the reasons given for the low abundance or absence of shrews in intensively managed orchards must be correct and not influenced by trapping.

## 5. Conclusions

Based on the results of the small mammal survey in agricultural and commensal habitats, it can be concluded that Soricidae (*S. araneus* and *S. minutus*) were an under-represented group of small mammals in orchards and homesteads, their proportion being less than the average in the other habitats. The diets of these two species in both habitat groups were similar, as determined by stable isotope analysis. We hypothesize that the main reasons for their limited abundance are intensive agricultural practices, contamination with plant protection products, and a lack of invertebrates, which are the main food of shrews. The presence of *N. fodiens* and *N. anomalus* was not expected in the homesteads, and their diet requires further investigation.

## Figures and Tables

**Figure 1 animals-13-01028-f001:**
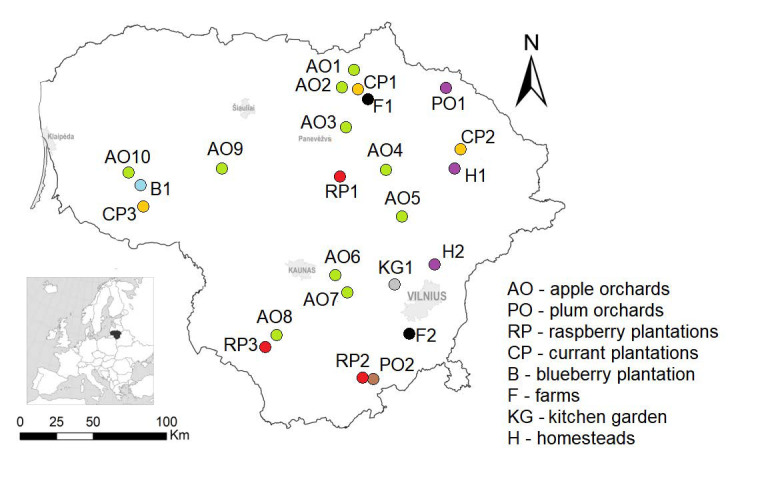
Study sites in Lithuania, 2018–2022.

**Figure 2 animals-13-01028-f002:**
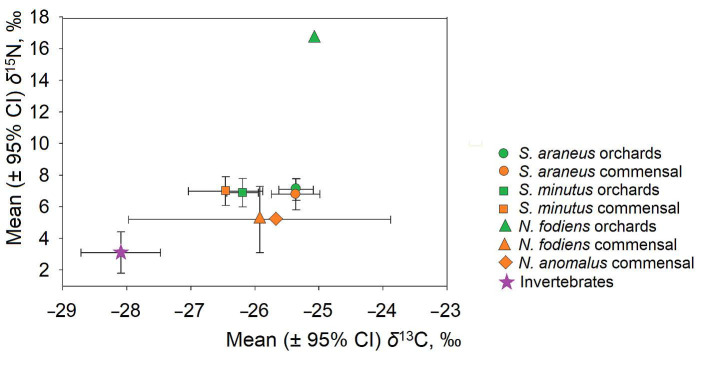
The position of insectivorous species and their possible prey in isotopic space according to stable isotope ratios of hair samples from commercial fruit farms and commensal habitats.

**Figure 3 animals-13-01028-f003:**
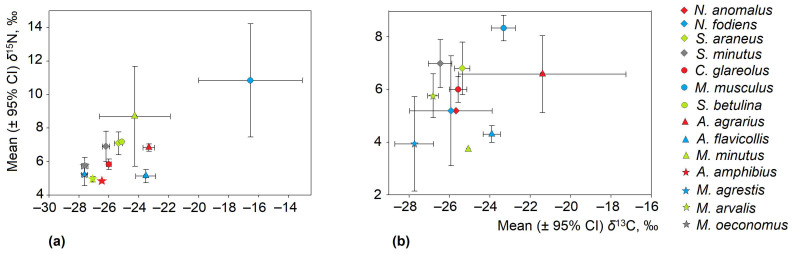
Isotopic position of Soricidae species in relation to other small mammal species based on stable isotope ratios in hair samples from commercial fruit and berry farms (**a**), and commensal habitats (**b**): △—granivores, ◯—omnivores, ✩—herbivores, ◊—insectivores. The rodent sample sizes in commercial orchards and commensal habitats were: *C. glareolus* 232/393, *M. musculus* 11/95, *S. betulina* 2/0, *A. agrarius* 430/213, *A. flavicollis* 540/322, *M. minutus* 18/1, *A. amphibius* 1/0*, M. agrestis* 42/20, *M. arvalis* 556/109, and *M. oeconomus* 46/14 individuals, respectively.

**Table 1 animals-13-01028-t001:** Characteristics of small mammal communities: N—number of trapped small mammal individuals; P%—proportion of all Soricidae species (95% CI); n1—number of *S. araneus*; n2—number of *S. minutus*; RA1—relative abundance of *S. araneus*, RA2—relative abundance of *S. minutus* (individuals per 100 trap days). Superscript letters denote differences, significant at *p <* 0.05.

Habitat	Year	N	P% (CI)	n1	n2	RA1	RA2
Control	2018–2022	830	4.9 ^a^ (3.6–6.6)	29	11	0.24 ± 0.06 ^a^	0.10 ± 0.05 ^a^
Orchard	2018–2022	804	0.8 ^b^ (0.3–1.6)	6	3	0.06 ± 0.03 ^b^	0.02 ± 0.01 ^a^
Plantation	2018–2022	302	2.7 ^b^ (0.2–5.0)	7	1	0.14 ± 0.06 ^b^	0.01 ± 0.01 ^a^
Kitchen garden	2019–2022	361	1.7 ^b^ (0.6–3.6)	3	2	n/a	n/a
Homestead	2019–2022	945	5.0 ^a^ (3.4–6.9)	20	9	n/a	n/a
Farm	2018, 2021–2022	199	0	0	0	n/a	n/a

**Table 2 animals-13-01028-t002:** Central positions and ranges of stable isotope ratios in the hair of small mammals from commensal habitats, 2019–2022.

Species	*δ*^13^C Values, ‰			*δ*^15^N Values, ‰		
	*n*	Mean ± SE	Range	*n*	Mean ± SE	Range
Commercial orchards						
*S. araneus*	24	–25.36 ± 0.13	2.61	24	7.08 ± 0.33	6.88
*S. minutus*	13	–26.19 ± 0.11	1.38	13	6.89 ± 0.42	4.58
*N. fodiens*	1	25.07	–	1	16.61	–
Kitchen garden						
*S. araneus*	1	–24.65	–	1	6.76	–
*S. minutus*	2	–27.47 ± 0.32	0.64	2	7.01 ± 0.58	1.16
*N. fodiens*	1	–26.10	–	1	5.02	–
Homestead						
*S. araneus*	11	–25.43 ± 0.17	1.97	11	6.80 ± 0.50	5.56
*S. minutus*	8	–26.21 ± 0.24	1.91	8	6.98 ± 0.50	4.15
*N. fodiens*	1	–25.76	–	1	5.36	–
*N. anomalus*	1	–25.67	–	1	5.19	–

## Data Availability

Due to ongoing investigation and preparation of PhD thesis, the data of this study are available from the second author upon reasonable request.

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
