# Peer review of "Shrews Under-Represented in Fruit Farms and Homesteads"

_animals, 2023, doi:10.3390/ani13061028_

Round 1
Reviewer 1 Report
I have read with great interest, the robust and important work of the authors, presented in the manuscript. I believe it is a work with important results and a lot of significant field work, including as well important analysis especially in respect to isotopic analyses.
I would suggest to the authors, the following comments to be taken into consideration for the improvement of the text.
Linguistic corrections
Line 40
Maybe write in a better and more understandable way, the phrase “including in its genetic diversity”. As it is now, it does not add well with the rest of the phrase, leaving the meaning of the phrase a bit blurry. Maybe remove “in”?
Lines 43, 61, 63
Maybe add the author name before the brackets of the citation [6]? Please consult as well with the Journal guidelines on that.
Similarly in Line 61 the citation [17]
Similarly in Line 63 the citation [19]
If this is not necessary according to guidelines, skip the comment.
Line 52
“is caught” could probably be written a bit better, maybe written as “trapped”, or “found”, “encountered”, etc.
The word “caught” is also used in other parts of the text (e.g. line 61 in the beginning). Although it is ok, I find it is not very suitable always to describe in a scientific context the meaning of trapped, sampled, etc. Maybe caught could be used in less points of the text alternated with the words suggested.
Concept and experimental design corrections
Lines 109, 123, 164-173
There are two references to stable isotopic analyses (which were used in order to examine the species trophic niche) in the “Introduction” section (lines 109 and 123), and a paragraph of 8 lines in “Materials Methods” section, in respect to isotope analysis (lines 165-173).
I believe, that although in line 169 it is mentioned that “more details in respect to sample preparation and analyses are given” in respect to reference [51], it would be better and advisable, to include a higher number of information concerning how one can achieve analysis of trophic niche through the use of stable isotope analysis, within the current manuscript.
I would suggest to include the basics of the process, and somehow mention/include here, the main axis of the “isotope analysis concept” and how it leads to effectively analyze the trophic niche, and why is it chosen as a method, if it is effective, why it is preferred and probably more effective than other methods.
Lines 126-143
As I understand, there are 23 different geographical sites. Nonetheless, the independent variables formulating the data analysis concept, have on one hand different spatial representation per site, but also different use in the data analyses. Therefore, I believe that additional explanations, probably with tables or a new map(s), should also present and better explain in the Material Methods, the different categories taken under consideration as factors -independent variables, and their levels. Maybe even Figure 1 should change or be divided in two maps. Let me elaborate on that.
I understand the authors have used the below levels of independent factors in their study:
1) Agricultural Sites with Habitat categories such as (i) Orchards (sub-categories: Apple and Plum), (ii) Plantations (sub-categories: Raspberry, Currant, Highbush)
2) Non-Agricultural Sites, or else Commensal Sites, with categories (i) Homesteads, (ii) Kitchen Garden, (iii) Farms.
3) Age of Habitat Types such as (i) Old Fruit Orchards, (ii) Middle Aged Fruit Orchards & Plantations (iii) Young Orchards & Plantations
4) Intensity of Agricultural Measures such as (i) High, (ii) Medium, (iii) Low
5) Control sites such as Meadows (sub-categories: Mowed, Unmowed)
I believe, that the map of Figure 1 should probably mere reflect the different 23 geographical sites sampled, or, to show the first level of different independent factors, for instance Agricultural Sites, Commensal Sites. Then, the map figuring now as Figure 1 could be used as Figure 2 to demonstrate the rest of the levels of the independent factors.
An additional table should also be included demonstrating the other levels, to clarify more the situation. For instance a Table presenting as Lines the 23 geographical sites, and as columns the (1) Agricultural Sites Orchards, (2) Agricultural Sites Plantations, (3) Commensal Sites, (4) Age of Agricultural Sites, (5) Intensity of Agricultural Measures, (6) Control Sites (there are further comments for the control sites). Within cells of this Table, the authors could also indicate the different sub-categories of each column e.g. young-middle-old age, homestead-kitchen-farm, high-medium-low agricultural measure, mowed-unmowed, etc. That will give a better conceived visualization which now can hardly be followed either in the text or through Figure 1.
Furthermore, the habitat sub-categories of Agricultural Sites such as Apple, Plum, Raspberry, Currant, Highbush, although they are presented both in text and in Figure 1, they only occupy a very minimum part of the text in results (lines 199-200) and they are nowhere mentioned in the discussion. It would be advisable that the authors could elaborate a bit more on the aspect of habitat effect on the studied small mammals.
Another important comment here, has to do with the use of Habitat Types as an independent factor in Line 132 “variety of habitats”. I believe the phrase “variety of habitats” is wrong, because what the authors actually look at here, is the “Age of the Cultivations” here. Habitat Type, would be correct when it is used to check the effect of Apple, Plum, Raspberry, Currant, Highbush, separately as different categories of Agricultural Sites as a habitat factor, upon the dependent variables. But this is not the case here, so I believe the phrase “variety of habitat types” should change possibly to “Age of Cultivations” which better reflects what the authors have included in this point of the analysis.
Furthermore, in respect to Habitat Types effect, I can only see one mention on “berry plantations” in Results lines 200 and 203, and of “apple orchards” in line 199 – I wonder what the results have also shown for the other habitat types. It should be explained clearly in the text the use of each (habitat type / age of cultivations) as data analysis factors, and be demonstrated respectively in Tables and within text in Material Methods and in Results as well, what are the findings even if there are no findings.
It would also be important that the authors explain a bit more what they mean when they mention high-medium-low level of agricultural intensity.
Line 136
In respect to the choice of control sites.
It is mentioned that control habitats, were mowed or unmowed meadows. It would be good if the author could explain, why these were chosen as control habitats. They were chosen as control habitats, due to the fact that they are not cultivated?
But then again, we see here again 2 levels of land use, mowed and unmowed meadows. This brings the question whether these meadows are natural, or, artificial agriculturally used. If the second case applies, then it would be good to know, why meadows are chosen as control sites since they also have a level of agricultural intensity measures, which in that case is the mowing of the land.
The basic question here is, whether meadows are chosen as control due to their not-agricultural character. But, if that is the case again, how do the authors treat the mowing-not mowing factor. Maybe, if the mowing/not-mowing is not implicated in any way, then meadows should be clarified that they are used as control sites as a whole, explaining why, and that the mowing/not-mowing factor is not implicated in the process of data analysis or affecting the control sites.
Line 141
It is confusing here that there is a mention of agricultural use within Commensal Sites. If Commensal Sites were used as non-agricultural independent factors, then this mention here complicates things. If these sites are not implicated in the data analysis as a factor with an agricultural aspect, then this reference should be omitted, or, be clarified that although there were agricultural uses within Commensal Sites these agricultural uses/sites were not sampled, and strictly Commensal Not Agricultural sites were trapped.
Line 146
How many lines/rows of 25 traps?
Line 148
It is mentioned that “trapping was also done around all accessible buildings”. This is somehow generic not giving the exact information of the trapping effort there.
Line 149
Although the authors write that more details are given in the references [49] and [50], I believe that the basics of the design should also somehow be explained here in the current manuscript, unless it can be argued why they are omitted, and why they are referenced as citations. I believe, an important part of the design explanation somehow is missing here.
Line 145-163
I believe a Table should be added here to clarify things. A table similar to the one suggested to be added before, where, in each different category of Independent factor used in the data analysis, the number of total trap nights will be demonstrated, and if necessary two tables as well, one with 23 geographical sites, different independent factors as columns and trap nights in each respective cell, and if the Temporal variable cannot fit here it should be explained in further in the text, or through another table.
As the text stands now, although it gives information it is a bit difficult to follow up.
As a general comment, I believe that the whole “2.2. Small Mammal Trapping” should be more clearly written, with more details.
Statistical Suggestions/questions
Lines 183-185
In respect to ANOVA as I understand it was applied in 3 different levels of independent variables: 1) trophic group (as assessed from isotope analysis according to my understanding), 2) age of orchards, 3) intensity of agricultural measures.
It would be great if the authors could also mention if their dataset meets all ANOVA assumptions, independency of samples/groups, normal distribution of dependent variables (mentioned in Line 187, but I believe no mention exists in results), homogeneity of variances, sample size.
General Comment on experimental design
It would also be great, if the authors could include some explanations, on how and if, the great heterogeneity of trap effort between habitats, agricultural/commensal, periods, affects somehow the data analysis or not.
Author Response
Rev#1 comments and answers
Comment: I have read with great interest, the robust and important work of the authors, presented in the manuscript. I believe it is a work with important results and a lot of significant field work, including as well important analysis especially in respect to isotopic analyses.
I would suggest to the authors, the following comments to be taken into consideration for the improvement of the text.
Answer: thank you for all comments, answers are given below point-to-point.
Linguistic corrections
Comment Line 40: Maybe write in a better and more understandable way, the phrase “including in its genetic diversity”. As it is now, it does not add well with the rest of the phrase, leaving the meaning of the phrase a bit blurry. Maybe remove “in”?
Answer: removed as advised.
Comment: Lines 43, 61, 63
Maybe add the author name before the brackets of the citation [6]? Please consult as well with the Journal guidelines on that.
Similarly in Line 61 the citation [17]
Similarly in Line 63 the citation [19]
If this is not necessary according to guidelines, skip the comment.
Answer: according the journal guidelines, references should be given by numbers in the square brackets.
Comment: Line 52 “is caught” could probably be written a bit better, maybe written as “trapped”, or “found”, “encountered”, etc.
The word “caught” is also used in other parts of the text (e.g. line 61 in the beginning). Although it is ok, I find it is not very suitable always to describe in a scientific context the meaning of trapped, sampled, etc. Maybe caught could be used in less points of the text alternated with the words suggested.
Answer: Line 54, changed to “encountered”, next three – to “trapped”
Concept and experimental design corrections
Comment: Lines 109, 123, 164-173
There are two references to stable isotopic analyses (which were used in order to examine the species trophic niche) in the “Introduction” section (lines 109 and 123), and a paragraph of 8 lines in “Materials Methods” section, in respect to isotope analysis (lines 165-173).
I believe, that although in line 169 it is mentioned that “more details in respect to sample preparation and analyses are given” in respect to reference [51], it would be better and advisable, to include a higher number of information concerning how one can achieve analysis of trophic niche through the use of stable isotope analysis, within the current manuscript.
I would suggest to include the basics of the process, and somehow mention/include here, the main axis of the “isotope analysis concept” and how it leads to effectively analyze the trophic niche, and why is it chosen as a method, if it is effective, why it is preferred and probably more effective than other methods.
Answer: the problem here is so-called “self-plagiarism” – information can be given once, then rephrased in the next publication. In the third manuscript, there are very limited options to present the same methods without being noticed. We, however, already analyzed granivores, herbivores, omnivore bank vole, omnivore house mouse, not talking about the publications concerning isotopic analyzes of small mammals from more natural habitats. So not much can be added without repetition. But still, to answer your comment, we extended chapter 2.3.
Comment Lines 126-143: As I understand, there are 23 different geographical sites. Nonetheless, the independent variables formulating the data analysis concept, have on one hand different spatial representation per site, but also different use in the data analyses. Therefore, I believe that additional explanations, probably with tables or a new map(s), should also present and better explain in the Material Methods, the different categories taken under consideration as factors -independent variables, and their levels. Maybe even Figure 1 should change or be divided in two maps. Let me elaborate on that.
I understand the authors have used the below levels of independent factors in their study:
1) Agricultural Sites with Habitat categories such as (i) Orchards (sub-categories: Apple and Plum), (ii) Plantations (sub-categories: Raspberry, Currant, Highbush)
2) Non-Agricultural Sites, or else Commensal Sites, with categories (i) Homesteads, (ii) Kitchen Garden, (iii) Farms.
3) Age of Habitat Types such as (i) Old Fruit Orchards, (ii) Middle Aged Fruit Orchards & Plantations (iii) Young Orchards & Plantations
4) Intensity of Agricultural Measures such as (i) High, (ii) Medium, (iii) Low
5) Control sites such as Meadows (sub-categories: Mowed, Unmowed)
I believe, that the map of Figure 1 should probably mere reflect the different 23 geographical sites sampled, or, to show the first level of different independent factors, for instance Agricultural Sites, Commensal Sites. Then, the map figuring now as Figure 1 could be used as Figure 2 to demonstrate the rest of the levels of the independent factors.
An additional table should also be included demonstrating the other levels, to clarify more the situation. For instance a Table presenting as Lines the 23 geographical sites, and as columns the (1) Agricultural Sites Orchards, (2) Agricultural Sites Plantations, (3) Commensal Sites, (4) Age of Agricultural Sites, (5) Intensity of Agricultural Measures, (6) Control Sites (there are further comments for the control sites). Within cells of this Table, the authors could also indicate the different sub-categories of each column e.g. young-middle-old age, homestead-kitchen-farm, high-medium-low agricultural measure, mowed-unmowed, etc. That will give a better conceived visualization which now can hardly be followed either in the text or through Figure 1.
Answer: you are perfectly right as for classification of the habitats in the study sites. To acknowledge your comment, we added Appendix A with the Table A1 containing necessary habitat information of the commercial orchards and berry plantations. Please note, that main level of the habitat is reflected in the Figure 1. We do not think,that visualization of these features as additional maps are necessary, as we do not analyze spatial elements of the shrew distribution, relative abundance or diets.
Comment: Furthermore, the habitat sub-categories of Agricultural Sites such as Apple, Plum, Raspberry, Currant, Highbush, although they are presented both in text and in Figure 1, they only occupy a very minimum part of the text in results (lines 199-200) and they are nowhere mentioned in the discussion. It would be advisable that the authors could elaborate a bit more on the aspect of habitat effect on the studied small mammals.
Answer: You are right, sub-categories of Agricultural Sites such as Apple, Plum, Raspberry, Currant, Highbush were used just in Lines 199–200, to present shrew proportion in small mammal communities. Sample size in the different crops was very small or zero, so we pooled data and used categories orchard/plantation further in the Table 1. As for Discussion, there are no data from the references to use for comparisons on such a level of habitat.
Comment: Another important comment here, has to do with the use of Habitat Types as an independent factor in Line 132 “variety of habitats”. I believe the phrase “variety of habitats” is wrong, because what the authors actually look at here, is the “Age of the Cultivations” here. Habitat Type, would be correct when it is used to check the effect of Apple, Plum, Raspberry, Currant, Highbush, separately as different categories of Agricultural Sites as a habitat factor, upon the dependent variables. But this is not the case here, so I believe the phrase “variety of habitat types” should change possibly to “Age of Cultivations” which better reflects what the authors have included in this point of the analysis.
Furthermore, in respect to Habitat Types effect, I can only see one mention on “berry plantations” in Results lines 200 and 203, and of “apple orchards” in line 199 – I wonder what the results have also shown for the other habitat types. It should be explained clearly in the text the use of each (habitat type / age of cultivations) as data analysis factors, and be demonstrated respectively in Tables and within text in Material Methods and in Results as well, what are the findings even if there are no findings.
Answer: we removed “variety of habitats”, as additional information is now presented in the Table A1.
Comment: It would also be important that the authors explain a bit more what they mean when they mention high-medium-low level of agricultural intensity.
Answer: We added text to explain intensity of agricultural measures: “Depending on soil scarification, grass mowing, mulching of the plant interlines, and the usage of rodenticides and plant protection agents, we characterized three levels of intensity. Sites with only grass mowing once or several times per season were attributed to low intensity, while usage of two measures from those above listed once or twice per season was defined as medium intensity. Application of several measures or frequent application of two measures per season was defined as high intensity.”
Comment Line 136 In respect to the choice of control sites. It is mentioned that control habitats, were mowed or unmowed meadows. It would be good if the author could explain, why these were chosen as control habitats. They were chosen as control habitats, due to the fact that they are not cultivated?
But then again, we see here again 2 levels of land use, mowed and unmowed meadows. This brings the question whether these meadows are natural, or, artificial agriculturally used. If the second case applies, then it would be good to know, why meadows are chosen as control sites since they also have a level of agricultural intensity measures, which in that case is the mowing of the land.
The basic question here is, whether meadows are chosen as control due to their not-agricultural character. But, if that is the case again, how do the authors treat the mowing-not mowing factor. Maybe, if the mowing/not-mowing is not implicated in any way, then meadows should be clarified that they are used as control sites as a whole, explaining why, and that the mowing/not-mowing factor is not implicated in the process of data analysis or affecting the control sites.
Answer: as a control, nearest non-orchard (non-plantation) and non-crop habitat was used. These were meadows or the forest. All these categories were pooled as “control”. So you are right, control habitats were non-agricultural. And, with small sample sizes, we do not analyze influence of the mowing/non-mowing. We now present data on the control habitats of every site in the Table A1.
Comment Line 141
It is confusing here that there is a mention of agricultural use within Commensal Sites. If Commensal Sites were used as non-agricultural independent factors, then this mention here complicates things. If these sites are not implicated in the data analysis as a factor with an agricultural aspect, then this reference should be omitted, or, be clarified that although there were agricultural uses within Commensal Sites these agricultural uses/sites were not sampled, and strictly Commensal Not Agricultural sites were trapped.
Answer: we excluded this, as in the analyses agricultural aspect of commensal habitats was not included (2 individuals of S. araneus and 1 individual of S. minutus were trapped in the garden of H1, with no shrews in other non-built areas in the other commensal sites).
Comment Line 146
How many lines/rows of 25 traps?
Answer: 2–4 lines in orchards and plantations, 1–2 lines in control habitats, depending on their size. Information added to the text. We cannot present elaborated details, as these are already published.
Comment Line 148
It is mentioned that “trapping was also done around all accessible buildings”. This is somehow generic not giving the exact information of the trapping effort there.
Answer: The same as inside the buildings, trapping effort outside was not regular, either. Explanatory text was added. We had no possibility to ensure equal trapping effort due to various owners and their voluntary schedules, therefore index of relative abundance is not known for commensal habitats.
Comment Line 149
Although the authors write that more details are given in the references [49] and [50], I believe that the basics of the design should also somehow be explained here in the current manuscript, unless it can be argued why they are omitted, and why they are referenced as citations. I believe, an important part of the design explanation somehow is missing here.
Answer: we added more information about trapping as per comment above. The main problem to keep it short with references to the previous publications is very simple – the same methods were already described, and cannot be replicated without a red mark on plagiarism check. We think this is not fair from the position of authors, but the methods cannot be described repeatedly in different words. Therefore, we are trying to avoid repetition using the references.
Comment Line 145-163
I believe a Table should be added here to clarify things. A table similar to the one suggested to be added before, where, in each different category of Independent factor used in the data analysis, the number of total trap nights will be demonstrated, and if necessary two tables as well, one with 23 geographical sites, different independent factors as columns and trap nights in each respective cell, and if the Temporal variable cannot fit here it should be explained in further in the text, or through another table.
As the text stands now, although it gives information it is a bit difficult to follow up.
As a general comment, I believe that the whole “2.2. Small Mammal Trapping” should be more clearly written, with more details.
Answer: to answer your comment, we added Table A2 with details of trapping effort in orchards and plantations. We do not see it is needed in the main text, as (i) details are given in the referenced publications, and (ii) with such a small number of shrews trapped, these differences did not affect results.
Statistical Suggestions/questions
Comment: Lines 183-185: In respect to ANOVA as I understand it was applied in 3 different levels of independent variables: 1) trophic group (as assessed from isotope analysis according to my understanding), 2) age of orchards, 3) intensity of agricultural measures. It would be great if the authors could also mention if their dataset meets all ANOVA assumptions, independency of samples/groups, normal distribution of dependent variables (mentioned in Line 187, but I believe no mention exists in results), homogeneity of variances, sample size.
Answer: we apologize mistype, the first level in your interpretation is habitat, not the trophic group. As for the dataset and ANOVA assumptions, we add Table A3 to show this information. In short, normality and homogeneity of variances was fulfilled for isotopic samples of both S. araneus and S. minutus.
To acknowledge limitations due to small sample size, at Line 189 text was added “the small sample size of two Neomys species may result in low power for statistical analysis”.
General Comment on experimental design: It would also be great, if the authors could include some explanations, on how and if, the great heterogeneity of trap effort between habitats, agricultural/commensal, periods, affects somehow the data analysis or not.
Answer: in the previous papers on other groups of small mammals we used rarefaction to test influences of different trapping effort, and none found. To acknowledge this comment, we added text “Differences in trapping effort were inevitable due to differences in habitat availability, but, as shown in our previous publications [49–51], they did not affect the trappability of the more abundant species and the diversity of the small mammal community.”

Reviewer 2 Report
The authors report on a study about the proportion of shrews in the small-mammal community of anthropogenic sites in Lithuania, and their diet using stable isotope analysis. The manuscript deals with a number of topics, most of which are covered only superficially, each if discussed in detail would be beyond of the scope of the framework, considering the very specific title.
My major concern regards the large number of killed animals in relation to meagre results. The researchers killed 3141 individuals from 14 species, but mention numbers and proportions only for shrews. The authors should at least have provided a corresponding table. To this, the snap-trapped rodents include members of protected species, such as Sicista betulina. I doubt that this is in accordance with European legislation, as stated by the authors.
In my opinion, removal trapping is an outdated method that should no longer be applied in scientific studies. Therefore I suggest to not accept the manuscript for publication. Still, I attached the pdf with my comments and corrections. I also highlighted some flaws in References.

Author Response
Rev#2 comments and answers
Comment: The authors report on a study about the proportion of shrews in the small-mammal community of anthropogenic sites in Lithuania, and their diet using stable isotope analysis. The manuscript deals with a number of topics, most of which are covered only superficially, each if discussed in detail would be beyond of the scope of the framework, considering the very specific title.
My major concern regards the large number of killed animals in relation to meagre results. The researchers killed 3141 individuals from 14 species, but mention numbers and proportions only for shrews. The authors should at least have provided a corresponding table. To this, the snap-trapped rodents include members of protected species, such as Sicista betulina. I doubt that this is in accordance with European legislation, as stated by the authors.
In my opinion, removal trapping is an outdated method that should no longer be applied in scientific studies. Therefore I suggest to not accept the manuscript for publication. Still, I attached the pdf with my comments and corrections. I also highlighted some flaws in References.
Answer:
Please have in mind that over 15 papers are published using this material, analyzing not only small mammals, but also their pathogens and reproduction parameters. None of these papers could be prepared without removal trapping and dissection. Some papers are underway – these also use internal organs and tissues of the snap-trapped animals.
In Lithuania, snap trapping of small mammals is perfectly legal. Moreover, we have approval from the Ethics Committee at the institution (shown in the Back matter), and we consulted several times with the Ministry of Environment. Their answer was that we do not require any permission. Moreover, landowners use rodenticides, snap traps, and even sticky mats to exterminate rodents on their properties.
Sicista betulina is the LC (least concern) species in Lithuania and neighboring Latvia. It is not protected. Two individuals were trapped in the non-characteristic habitat, unintentionally. As rodents are being poisoned with rodenticides in commercial orchards, these individuals anyway should be considered dead. So, our results add knowledge about the possibility of this species inhabiting agricultural habitats.
Thank you for understanding this. In fact, three or four times we discussed trapping with reviewers of our other publications, and all these were accepted after explanations.
As for the other comments, we are grateful for these, and proposed changes were all implemented.
AO1 is young orchard (line 134), we thank you very much for noticing this mistype.
We changed the text adding common names to species on the first mention (lines 252–258), and, combined with the comment of Rev#3, sample sizes for rodents are added into the Fig. 3 caption.
References were corrected according your comments.

Reviewer 3 Report
Review of ms animals-2234893, “Shrews under-represented in fruit farms and homesteads”.
This study investigates the spatial distribution of shrews in Lithuania from surveys done between 2018 and 2022 using snap-trapping. This study is of significance given that shrews could be valuable indicator species, though too rarely considered in animal studies. The authors show that shrews are under-represented in this sample. More importantly, these species were sampled at higher frequency in meadows and homesteads than in highly anthropized agroecosystems. The authors also investigate species diet using carbons/nitrogen stable isotope analysis.
I appreciate this study done on a poorly studied set of small mammal taxa. The paper is well presented and I believe will be suitable for publication in the journal Animals after some minor revision.
Find below my general comments:
- A minor comment is that the introduction gives extensive reviews and details. For the readers’ comprehension, it may somehow be reduced to better emphasize the objectives of this study and put it into context. However, this is a minor comment.
- Study design seems appropriate to me. However, the authors should emphasize in the text that small sample size for some of the study species may result in low power of the statistical analyses.
- Methods are adequately described, however the authors should clearly define how they use the term “commensal”, i.e. does it refer to inside human buildings (commensal sensu stricto) or also to their immediate outside surroundings (commensal sensu lato)? This needs clarification, see ambiguous use of this terms as shown in Table 2 (Line 228) or L 238. I would rather suggest to restrict the use of commensal term to inside buildings and their immediate surroundings.
- conclusions are supported by the results. But authors should be less conclusive about the impact of pesticides / invertebrate abundance on shrew frequency. Similarly, conclusions on the diet of Neomys should be less affirmative given the very low sample size for these taxa.
- The authors may explain why they analysed only variations among sampling sites, but not temporal variations.
- Trapping effort: should it be referred to as “trap*nights” rather than “trap days” throughout the text?
Find below suggestions in specific parts of the text:
- Line 9: “between 2018 and 2022” instead of “2018 and 2022”
- L 17: “may be factors limiting” instead of “may be may be a factor limiting”
- L 30-31: I would suggest to remove “and 3.03 ‰/13.50 ‰ for N. fodiens in the orchards, indicating the possibility that the latter species 30 was consuming not only invertebrates”. Do not put too much emphasis on a poorly supported result. Sample size is low (n=3 ind.) and this result rely only a single data with high delta 15N.
- L 32: “We propose that the” instead of “We presume the”
- L 79: “countries” instead of “counties”
- L 91: “maize and sunflower fields” instead of “maize, sunflower and sunflower fields”
- L 170: when referring to “invertebrates”, could the authors give more information (arthropods, earthworms, etc., both?)
- L 204-205: I may have missed something, but the trapping success in the text given for S. araneus (2.67/100 trap days) and S. minutus (4.00/100 trap days) seems to me different from values given in table 1. Please correct if necessary or explain this issue.
- L 207: give “(ANOVA, F = ?, p = ?”) for S. minutus
- L 225-226: specify that this result rely only on data from a single individual.
- L 232-233: specify that this result rely only on data from a single individual
- idem in L 326-328: specify that this result rely only on data from a single individual
- Figure 3: sample size for non-shrew small mammal species (rodents) should be given in the text or figure.
- Figure 3: y axis of Fig 3a and Fig 3b should be given at same scale for easier visual comparison
- L 298: “We propose the hypothesis” instead of “We believe”
- L 338: “exclude” is rather too much affirmative. Instead use “suggest” or “propose”.
Author Response
Rev#3 comments and answers
Review of ms animals-2234893, “Shrews under-represented in fruit farms and homesteads”.
This study investigates the spatial distribution of shrews in Lithuania from surveys done between 2018 and 2022 using snap-trapping. This study is of significance given that shrews could be valuable indicator species, though too rarely considered in animal studies. The authors show that shrews are under-represented in this sample. More importantly, these species were sampled at higher frequency in meadows and homesteads than in highly anthropized agroecosystems. The authors also investigate species diet using carbons/nitrogen stable isotope analysis. I appreciate this study done on a poorly studied set of small mammal taxa. The paper is well presented and I believe will be suitable for publication in the journal Animals after some minor revision.
Answer: thank you for your comments, we acknowledge most of them (see below), or explain the rebuttal.
Find below my general comments:
Comment: A minor comment is that the introduction gives extensive reviews and details. For the readers’ comprehension, it may somehow be reduced to better emphasize the objectives of this study and put it into context. However, this is a minor comment.
Answer: this group is under-represented in small mammal investigations from the region, therefore we presented more details about this group. Thank you for understanding.
Comment: Study design seems appropriate to me. However, the authors should emphasize in the text that small sample size for some of the study species may result in low power of the statistical analyses.
Answer: we added “the small sample size of two species may result in low power for statistical analysis.” to 2.4 chapter.
Comment: Methods are adequately described, however the authors should clearly define how they use the term “commensal”, i.e. does it refer to inside human buildings (commensal sensu stricto) or also to their immediate outside surroundings (commensal sensu lato)? This needs clarification, see ambiguous use of this terms as shown in Table 2 (Line 228) or L 238. I would rather suggest to restrict the use of commensal term to inside buildings and their immediate surroundings.
Answer: we added text with explanation, “Trapping was also done around all accessible buildings – thus, the term “commensal habitats” was used sensu lato”.
Comment: Conclusions are supported by the results. But authors should be less conclusive about the impact of pesticides / invertebrate abundance on shrew frequency. Similarly, conclusions on the diet of Neomys should be less affirmative given the very low sample size for these taxa.
Answer: thank you. We put “should be” instead of “are”. We did not have a conclusion on the Neomys diets, just mention “their diet needs further investigation”.
Comment: The authors may explain why they analysed only variations among sampling sites, but not temporal variations.
Answer: due to the low sample size of shrews, we analyse data pooled by habitats, not by sampling sites. Temporal variation cannot be analysed, as sample size in several years is zero, see Table 1 – even in pooled data it is very small.
Comment Trapping effort: should it be referred to as “trap*nights” rather than “trap days” throughout the text?
Answer: we use “trap days”, where the word “day” means “day AND night” – traps were kept for three consecutive days and nights without a break. In fact, usage or this term was influenced by previous reviewers – we used “trap days” instead of “trap nights” in Animals already.
Find below suggestions in specific parts of the text:
Comment Line 9: “between 2018 and 2022” instead of “2018 and 2022”
Answer: corrected as suggested, apologies for this
Comment L 17: “may be factors limiting” instead of “may be may be a factor limiting”
Answer: corrected as suggested
Comment L 30-31: I would suggest to remove “and 3.03 ‰/13.50 ‰ for N. fodiens in the orchards, indicating the possibility that the latter species was consuming not only invertebrates”. Do not put too much emphasis on a poorly supported result. Sample size is low (n=3 ind.) and this result rely only a single data with high delta 15N.
Answer: removed as suggested.
Comment L 32: “We propose that the” instead of “We presume the”
Answer: corrected as advised
Comment L 79: “countries” instead of “counties”
Answer: mistype corrected.
Comment L 91: “maize and sunflower fields” instead of “maize, sunflower and sunflower fields”
Answer: apologies, one of the crops was “corn”
Comment: L 170: when referring to “invertebrates”, could the authors give more information (arthropods, earthworms, etc., both?)
Answer: “Small molluscs and arthropods” inserted instead of “invertebrates”
Comment L 204-205: I may have missed something, but the trapping success in the text given for S. araneus (2.67/100 trap days) and S. minutus (4.00/100 trap days) seems to me different from values given in table 1. Please correct if necessary or explain this issue.
Answer: as it is written, “The HIGHEST relative abundance” in the text is really not the same as average values in the Table. No correction needed.
Comment L 207: give “(ANOVA, F = ?, p = ?”) for S. minutus
Answer: inserted, (ANOVA, F2,322 = 0.92, p = 0.43)
Comments: L 225-226: specify that this result rely only on data from a single individual; L 232-233: specify that this result rely only on data from a single individual; idem in L 326-328: specify that this result rely only on data from a single individual
Answer: repeating of values is not advised by the journal, sample sizes are shown in the Table 2, therefore we cannot repeat these in the text.
Comment Figure 3: sample size for non-shrew small mammal species (rodents) should be given in the text or figure.
Answer: As sample sizes of shrews were already given in the Table, these cannot be repeated in the legend of Figure 3. In the parts of Fig. 3 (a) and (b) sample sizes differ, requiring double legend. Therefore, we find best possibility to show rodent sample sizes in the figure caption.
Comment Figure 3: y axis of Fig 3a and Fig 3b should be given at same scale for easier visual comparison
Answer: we originally did parts (a) and (b) at the same scale, and then part (b) is so cluttered, that intervals greatly overlap. It was no way easier to see part (b) in this way.
Comment L 298: “We propose the hypothesis” instead of “We believe”
Answer: change done as proposed
Comment L 338: “exclude” is rather too much affirmative. Instead use “suggest” or “propose”.
Answer: text changed as “Finally, we suggest that there was no possible influence”

Round 2
Reviewer 1 Report
I would like to thank the authors for their effort in reshaping their manuscript, it is much appreciated.
In respect to the reviewed version of the authors' manuscript, I would like to kindly suggest though, the following issues, as a few more points are risen now that more details are presented on the text.
1.
In respect to my comment for Lines 43, 61, 63, I believe that what I wrote to the authors in first place, possibly wasn' t very clear. I did not ask, whether references should be given within brackets or not, since that is a clear guideline of the Journal.
I rather mostly asked, on whether the authors should write e.g.:
According to Linas [54], or,
According to [54].
The authors use the 2nd choice in 2-3 occasions in the text, but to my understanding, it would seem more appropriate to use the 1st. Please consult with the editorial for the best referencing in such occasions.
2.
In respect to the authors' answer concerning the Lines 109, 123, 164-173 and my request on presenting more methodological analysis in respect to the Isotope Analysis.
I understand what the authors state, about "self plagiarism". On the other hand, I firmly believe it is not possible for any reader or reviewer, to have to go to other publications in order to read the exact context of the methodological approach, on any step of the experiment design. As such, different redacting-ways must be found in order for manuscripts to be "stand-alone", in the information they provide, to readers.
At the moment, the Section 2.3 is certainly more understandable as it has been enhanced by the authors.
I would suggest though to the authors, to add one more bit of information in 2.3 Section, which I find necessary to exist within the text: Independently on whether we know it or not, it must be explained, what is the purpose of using isotope analysis in general, and why the authors use it here in their work. The context and essence/goal of using isotope analysis. After all, Material & Methods section is to explain what we do and why/how we do it.
3. Important Comments about Methodology and Data Analysis
Now that the authors have included more details about the trapping efforts per site, habitat type, age of habitats, etc., I must admit I am sceptic about the methodological focus of the article. The more I read the article, the more it seems that it's focus is mixed, leaving many points vaguely explained.
I firmly believe the authors should really, try to clarify the following in order to reach a clear-methodological-goal and robuts manuscript for publication in Animals:
3.1
In respect to experimental design.
In Table A2 a huge difference appears between the trapping effort in Apple Orchards and in other habitat types. Furthermore:
If I undestand well, from the manuscripts it is deduced that in commensal habitat types the trapping effort was 1.530+210+1480 = 3.220 trap nights.
Whereas, as it derives from the Table A2, the trapping effort to all the other habitat types was actually a total of 36.828 trap nights.
IF that is the case, then it seems like this is a highly non-balanced experimental design, where the Commensal Habitats are under-sampled. In an occasion where 3.220 trap nights are offered to Commensal Habitats (almost 8% of all trapping effort) versus to the 36.828 trap nights offered to all other habitat types of Table A2 (almost 92% of all trapping effort), then maybe it is due to the initial experimental design and varied trapping effort that shrews are under-represented in Commensal Habitats. If this varied trapping effort between habitats and this initial experimental design, does not affect the statistical analysis with bias, it would be great to be explained by the authors in the manuscript.
3.2
In respect to experimental design.
Considering this high difference in trapping effort between habitat types, it would be great to clarify in the manuscript, if the authors could present explanations of statistical/mathematical approaches on dissimilar trapping efforts in different habitats (as it happens on their study), and how they are treated in order to reach meaningful conclusions in other studies, focusing though on the mathematical/statistical principels. For instance, they could bring references to the text of how statistical approaches are applied on different sample sizes and how these samples are treated according to statistical basis, and why, and upon which mathematical principles they are applied, in order to reach robust conlusions after the data analysis.
Although the authors claim that "Differences in trapping effort were inevitable due to differences in habitat availability, but, as shown in our previous publications [49–51], they did not affect the trappability of the more abundant species and the diversity of the small mammal community", I believe it not that much a question of trappability of abundant species, but mostly on whether statistical tests can be applied on such dissimilar trapping efforts and reach robust conclusions, and, in that context, what happens with not-abundant species.
3.3
In respect to experimental design.
There is a conflict that I locate, between the "Agricultural Intensity Factor" and the Control Habitats "Mowed / Unmowed Meadows".
As it is deduced from the Table A1, and the manuscript text, and the author answers to the 1st Review, Meadows are chosen as control habitats because they are not agricultural habitats.
There is a contradiction here though:
As a level of agricultural practices and Agricultural Intensity, "Grass Mowing" is considered clearly a level of Agricultural Intensity in agricultural habitats.
In the Control Habitats though (the Meadows), Mowing which also recorded from the authors, but it does not seem to be included as an Agricultural Practice. Why is that?
To my understanding, if "Mowing" is a level of Agricultural Intensity factor in other habitats, it should also be included as a level of Agricultural Intensity in "Mowed Meadows" - it probably also affects small mammal communities and species presence/trapping.
And in that case, then probably, the "Mowed Meadows" should be a part of Agricultural habitats, or habitats with Low Agricultural Intensity, or to be excluded from the analysis, and possibly only the Non-Mowed Meadows should be the clearly Control - Non Agricultural Habitats.
It that is the case, then, the total representation of the results and data analysis should be restructured.
The authors claim in their response that "with small sample sizes, we do not analyze influence of the mowing/non-mowing". Stil, it is not correct to treat it as such, and furthermore, according to Table A2 "Mowed Meadows" they have a trapping effort of 7.510 trap nights which is the second highest of the 8 habitat types, so I do not exactly understant what the authors mean when they state "small sample sizes". It appears that they have a very large trapping effort sample size in Mowed Meadows.
4
General Comments
4.1
I would like to kindly suggest, that the authors focus their paper on the Isotope Analysis facts, and remove from their manuscript the assessment of small mammal community differences between different habitats, habitat ages, Commensal / Agricultural habitats. Unless they can very clearly explain the different steps in respect to Independent Variables/Factors - Dependent Variables/Small mammal presence that they have followed. Right now, it still remains difficult to follow up.
4.2
In addition to point 4.1, in many occasions, the authors claim that they have published in other publications the details that I have asked them to clarify within their manuscript. Nonetheless, I firmly believe that it is highly difficult to have a "generic" context in a stand-alone new publication, because most of the info has already been published elsewhere, in order to just avoid plagiarism. That also brings me, as well, to my above point 4.1.
4.3
Due to the many differences between trapping effort / sample size / possible statistical bias / appropriate explanations of the accuracy of the statistical approaches in each occasion, it would be great to bring into the manuscript explanations on how this varied sample sizes and different trapping efforts are treated from a statistical/mathematical point of view.
Author Response
Please find answers attached

Reviewer 2 Report
Although I still object against snap trapping, I understand the authors' arguments listed in their answer.
Author Response
Rev#2
Comments and Suggestions for Authors
Comment: Although I still object against snap trapping, I understand the authors' arguments listed in their answer.
Answer: thank you for your understanding